# Omega-3 Fatty Acid Consumption Alters Uterine Contraction: A Comparative Study on Different Breeds of Rats

**DOI:** 10.3390/ijms26115221

**Published:** 2025-05-29

**Authors:** Kalman F. Szucs, Dora Vigh, Seyedmohsen Mirdamadi, Reza Samavati, Annamaria Schaffer, Tamara Barna, Tamás Tóth, György Bázár, Henrik Baranyay, Robert Gaspar

**Affiliations:** 1Department of Pharmacology and Pharmacotherapy, Albert Szent-Györgyi Medical School, University of Szeged, 6725 Szeged, Hungary; szucs.kalman@med.u-szeged.hu (K.F.S.); vigh.doraa@gmail.com (D.V.); moh.mirdamadi@gmail.com (S.M.); reza.samavat@gmail.com (R.S.); ann.schaffer94@gmail.com (A.S.); b.tamara.9509@gmail.com (T.B.); 2Agricultural and Food Research Centre, Széchenyi István University, 9026 Győr, Hungary; toth.tamas@ke.hu; 3Adexgo Ltd., 9021 Győr, Hungary; george.bazar@adexgo.hu; 4UBM Feed Ltd., 2085 Pilisvörösvár, Hungary; henrik.baranyay@ubm.hu

**Keywords:** PUFA, fatty acid, EPA, DHA, breed-dependent, contractility, sex hormone, uterus, rat, smooth muscle electromyography

## Abstract

Polyunsaturated fatty acids (PUFAs) play roles in several physiological and pathophysiological processes, but their effects on reproductive function are controversial. The aim of the study was to investigate the effect of n-3 PUFA-rich fish oil and n-6-rich sunflower oil on sex hormone status, in vivo and in vitro uterine contractility, and endometrial remodeling. Female Sprague Dawley, Lister hooded, and Wistar rats were treated orally for 20 days with 1 mL of tap water, sunflower oil, or fish oil. Blood samples were taken for gonadotropic and sex hormone analysis. In vivo smooth muscle contractions were measured weekly by electromyography. Isolated uterine and cecal contractions were measured after sacrificing the animals. Endometrial remodeling was detected based on the presence of αvβ3 integrin by optical imaging. In Sprague Dawley rats, fish oil increased the LH level and progesterone/estradiol (P4/E2) ratio compared to the sunflower oil-treated group. Uterine contractions were reduced both in vitro and in vivo. Endometrial αvβ3 integrin activity was increased in the fish oil group. In Lister hooded rats, neither sunflower nor fish oil treatments modified the investigated parameters. However, in Wistar rats, both oils increased only the in vivo contractions and reduced the P4/E2 ratio, along with αvβ3 integrin fluorescence. n-3 PUFA-rich fish oil induces a breed-dependent effect on sex hormone status and uterine contractions in rats. The response to PUFA intake may vary significantly within a given species, which may have importance both in animal feeding and human nutrition.

## 1. Introduction

Fatty acids are fundamental constituents of the lipid bilayers that make up the cellular membrane. Furthermore, they are regarded as a crucial energy source, contributing to an estimated 30% of overall energy expenditure [1]. From a chemical perspective, fatty acids are classified into distinct categories, such as n-3, n-6, and n-9, depending on the location of the initial double bond. These polyunsaturated fatty acids (PUFAs) are non-convertible and exhibit markedly different biochemical functions, the investigation of which has received significant attention in recent decades [2]. The predominant and most prevalent n-3 PUFAs include α-linolenic acid (ALA), stearidonic acid, eicosapentaenoic acid (EPA), docosapentaenoic acid, and docosahexaenoic acid (DHA). n-3 PUFAs are classified as essential fatty acids due to the limited capacity of mammalian cells to synthesize EPA and DHA from the precursor ALA. Consequently, these fatty acids must be obtained through dietary sources, such as salmon, fish oil, and specific algae species, as they are crucial for improving health and reducing the risk of various diseases [3,4]. However, in the westernized human population, due to the increased ingestion of n-6-fatty plant-based oils and n-6-fatty meat from animals fed n-6-fatty cereals, dysregulation and an increase in the ratio of n-6 to n-3 can be observed, which leads to a number of diseases [5]. The human nutritional regimen is characterized by a PUFA ratio of n-6 to n-3 of 1:1; however, contemporary dietary patterns have increased this ratio to a range between 10:1 and 25:1 [6].

The advantageous effects of n-3 PUFAs on individuals with cardiovascular diseases have been thoroughly investigated [2]. Their antiarrhythmic, antithrombotic, and hypotensive properties have been proven, along with their beneficial impact on inflammatory diseases (e.g., arthritis, psoriasis, and ulcerative colitis). n-3 PUFAs also induce cell death through the initiation of ferroptosis in various cancer cell lines [7,8] and may contribute to anxiety reduction and relief of depressive symptoms, as well as to improvements in cognitive function and a decreased risk of dementia [1,9].

n-3 PUFAs have multiple mechanisms of action; they enhance membrane fluidity, influence membrane elasticity, modulate intracellular signaling pathways, regulate gene expression, and facilitate the synthesis of eicosanoids, including prostaglandins (PG), prostacyclin, thromboxane, leukotrienes, and lipoxins. Arachidonic acid (AA), classified as an n-6 PUFA, is metabolized into various eicosanoids, including PGs and thromboxanes, which induce various pro-inflammatory responses. In contrast, n-3 PUFAs are metabolized in a position-specific manner to generate a different series of mediators that exhibit anti-inflammatory properties. Substituting AA by EPA and DHA in membrane phospholipids considerably alters the inflammatory response by decreasing the production of TNF-α, IL-6, and IL-1β in almost all cell types. The cyclooxygenase-1 (COX-1) enzyme is directly inhibited by DHA [9,10]. COX-2 has been shown to convert EPA to PGH_3_, while COX-1 cannot recognize EPA as a substrate. These findings suggest that the action of COX-1-derived PGs could be disrupted by n-3 PUFAs [11].

New findings suggest that women with dysmenorrhoea may experience less pelvic pain when including n-3 PUFAs in their diet. Diets abundant in n-6 PUFAs have been shown to markedly improve the phospholipid composition of cellular membranes. During the menstrual cycle, as progesterone levels fall, n-6 PUFAs, notably AA, are released and enter the PG synthesis pathway, promoting the delivery of PGs and leukotrienes to the uterus and increasing inflammation, which causes symptoms such as pain, nausea, vomiting, swelling, and headaches. The formation of PGF_2α_ and COX from AA triggers vasoconstriction and uterine contractions, which can result in ischemia and pain [12,13,14].

An irregularity in the balance of cytokines can lead to inflammatory reactions, contributing to negative pregnancy outcomes such as abortion, preterm birth, and intrauterine growth restriction [15]. Activities related to labor stimulation, uterine contraction, and cervical ripening during late gestation are mediated by pro-inflammatory cytokines [16]. Taking n-3 PUFAs, particularly DHA, as a supplement has been associated with a slight increase in the gestational period and newborn weight; however, it does not affect the probability of preterm birth or low weight at birth, and there is no consistent evidence to support benefits for maternal or infant health during the peripartum phase. The perinatal implications of the supplementation of n-3 PUFAs have also been shown to be inconsistent [17,18].

DHA also modifies the lipid composition of the membranes in pregnant human myometrial smooth muscle cells, specifically PHM1-41. PUFAs have been found to inhibit the mobilization of [Ca^2+^]i induced by oxytocin in a cell culture model. These observations, along with the established capacity of dietary PUFAs to modify lipid composition, provide new insight into potential mechanisms associated with documented cases of dietary fish oil intake prolonging gestation [19].

It is known that in young women of childbearing age, the use of various nutritional supplements that also contain PUFA is widespread. Since limited data are available on how PUFAs directly modify uterine contractility and endometrial functions, our aim was to perform a comparative preclinical study to investigate the effects of n-3 PUFA-rich fish oil and n-6-rich sunflower oil supplementation on gonadotropic and sex hormone status, as well as in vivo contraction and endometrial remodeling in three different breeds of rats. New methodological approaches such as smooth muscle electromyography (SMEMG) and in vivo imaging system (IVIS) techniques have been used to measure breed-specific responses in order to detect possible differences.

## 2. Results

The body weight of the rats typically corresponded to their development according to their age at the time of inclusion in the experiment. The 20-day oral treatment did not cause a significant difference in the growth of rats in the case of sunflower oil or fish oil within the given breed compared to the control group (Figure 1).

### 2.1. Gonadotropic and Sex Hormone Levels

The 20-day treatment with fish oil significantly increased luteinizing hormone (LH) levels in the Sprague Dawley rats compared to the control group; however, there was no significant change in the Lister hooded rats. On the contrary, in the case of the Wistar rat breed, 20 days of sunflower oil treatment significantly reduced the plasma LH level of rats (Figure 2).

The ratio of P4/E2 hormones was also significantly changed by different oil treatments (Figure 3). Similar to the alteration in LH plasma concentration, 20 days of fish oil therapy significantly increased the relative P4/E2 ratio in Sprague Dawley rats; however, in the case of the Wistar rats, a significant decrease in the P4/E2 ratio was observed in both fish oil and sunflower oil-treated rats.

### 2.2. In Vivo Contractility

Smooth muscle myoelectric activity was detected in anesthetized rats and expressed as the power spectrum density maximum (PsD_max_) in the frequency range of 1–3 cycles per minute (cpm). In Sprague Dawley rats, fish oil treatment significantly reduced the PsD_max_ values, which varied in proportion to the strength of contractions compared to the control group. On the other hand, in the case of the Wistar rats, both sunflower oil and fish oil induced a significant increase in smooth muscle contractions compared to the control group of rats. However, there was no significant change in the Lister hooded rats as a result of treatments (Figure 4).

### 2.3. In Vitro Contractility

Smooth muscle contractions were detected in our isolated organ bath study. In Sprague Dawley rats, fish oil treatment significantly reduced both uterine and cecal contractions compared to the control and sunflower oil-treated group. On the other hand, in the case of the Wistar rats, sunflower oil induced a significant increase in the contractions of both organs compared to the control area under the curve (AUC) values. However, there was no significant change in the Lister hooded rats as a result of the treatments (Figure 5).

KCl-elicited contractions were also examined in the non-pregnant uterus and cecum. In contrast to the spontaneous response, oil treatments significantly increased KCl-evoked contractions in sunflower oil-treated Wistar rats, while they did not modify activity in the other two breeds (Figure 6).

### 2.4. Uterine Activity of αvβ3 Integrin

The intensity of αvβ3 integrin staining of the non-pregnant uterine horns was measured using the IVIS imaging procedure. The effect of the 20-day oil treatment was quantified after the regions of interest (ROI) evaluation of the raw images (Figure 7).

The fluorescent probe showed a significantly increased intensity in fish oil-treated Sprague Dawley rats compared to the control group, indicating increased αvβ3 integrin activity. However, treatment with sunflower and fish oil significantly reduced the intensity of αvβ3 integrin in the non-pregnant uterus in the Wistar rats compared to the control group. During our imaging study, there was no significant change as a result of oil treatments in the case of the Lister hooded rats (Figure 8).

## 3. Discussion

Although the effects of PUFAs have been extensively studied in both animals and humans, limited data are available on their effects on visceral smooth muscle contraction. In addition, there are no data on their breed-specific effects in the same species. The three breeds used in our investigation are within the same species of *Rattus Norvegicus*. The Sprague Dawley and Wistar rat breeds exhibit a close phylogenetic relationship and share a common ancestor; conversely, the lineage of Lister hooded rats remains ambiguous [20]. Our findings have shown that a 20-day administration of n-3 or n-6 fatty acids elicited breed-dependent effects on uterine functions in rats. In all breeds, the direction and magnitude of the in vitro contractility responses of the uterus and the cecum were the same, so the change in SMEMG measured in the frequency range 1–3 cpm always included the response of the cecum.

In the case of Sprague Dawley rats, the treatment with n-6-rich sunflower oil did not induce significant alterations in any of the measurement methodologies compared to the control group. However, the data unequivocally show that a higher intake of n-3 fatty acids, facilitated by fish oil supplementation, markedly decreases the spontaneous contractility of both the uterus and the cecum, as evidenced by in vivo SMEMG, and this reduction is corroborated by findings from isolated organ contractility assessments. This phenomenon is presumably attributable to the markedly elevated plasma ratio of P4/E2, as it is well established that an increased progesterone ratio corresponds to a decrease in contractile activity [21]. The relative increase in progesterone levels may have been initiated by an increase in the plasma concentration of LH through its physiological effects. All of these alterations are easily observable in the increased activity of the uterine αvβ3 integrin after fish oil treatment, suggesting that the 20-day fish oil regimen advanced rats into the proestrus-estrus phase of their reproductive cycle. It has been described previously that LH and FSH indirectly affect uterine αvβ3 integrin levels via their effect on sex hormone secretion. The expression of αvβ3 under the influence of ovarian steroids has shown that its regulation is mainly dominated by progesterone. Although estrogen acts as a major inhibitor, high concentrations of progesterone correlate not only with the presence of αvβ3 integrin but also with the expression of other integrins that contribute to endometrial receptivity to implantation, which presumably enhances animal fertility and the retention of pregnancy [22,23,24,25,26].

The metabolic implications of PUFAs on the progesterone profile throughout the estrus cycle and their involvement in the pathogenesis of polycystic ovary syndrome (PCOS) are also controversial [6]. Lu et al. found that the elevated dietary intake of EPA and DHA is inversely associated with the prevalence of PCOS and positively correlated with follicle-stimulating hormone (FSH), LH, and sex hormone-binding globulin (SHBG) [27]. Conversely, some investigations have reported no significant associations between n-3 PUFAs and the levels of FSH, LH, and SHBG [28]. Furthermore, in a study involving ovariectomized rats, the administration of EPA and DHA increased the expression of estrogen receptor-α [29].

Unlike Sprague Dawley rats, the Lister hooded rat breed exhibited no significant changes in any of the measured parameters after the 20-day fish oil treatment, nor did sunflower oil treatment produce any significant differences compared to the control group. However, the baseline values of the three rat breeds were consistent, making the ineffectiveness of the oil treatments incongruent with potential differences in fundamental physiological parameters.

In the case of Wistar rats, the use of sunflower oil and fish oil contributed to an improvement in the contractions of the isolated uterine and cecal samples. This increase in contractility was observed during in vivo SMEMG. This phenomenon was likely attributable to a reduction in LH levels in the rats following oil treatment, which subsequently lowered the P4/E2 ratio, thereby physiologically facilitating an increase in contractility and a decrease in αvβ3 integrin expression. The reaction of Wistar rats to sunflower oil treatment is in sharp contrast to the alterations observed in Sprague Dawley rats after fish oil treatment. In the case of Wistar rats, oil treatment, regardless of the proportions of n-3 and n-6 fatty acids, resulted in a decrease in the P4/E2 ratio and an increase in in vivo smooth muscle contractions, which substantially undermines the probability of successful gestation.

Research on rodent models shows that an overabundance of n-3 PUFAs changes uterine smooth muscle contractions. A variety of mechanisms have been proposed to elucidate these phenomena. These effects may arise directly from alterations in the biosynthesis of PGF_2α_ and PGE_2_, which serve to induce and alleviate myometrial contractions, respectively [11,30,31]. It is crucial to recognize that modifications in fatty acid composition can also affect contractile function through pathways distinct from PG synthesis. One conceivable mechanism involves the direct influence of fatty acids on the intrauterine expression of pivotal proteins associated with contraction. Such a protein is, for example, connexin-43, which is indispensable for the establishment of gap junctions that function as intracellular channels to promote electrocoupling and the intercellular transfer of myoelectric pulses. The smooth muscle that makes up the functional unit is distinguished by its pacemaker activity; the SMEMG technique represents a commonly used methodology for the detection of its slow wave signals [32,33].

We hypothesize that breed-specific responses are due to their possible different metabolizing properties, as was proven in the case of drugs or a high-fat diet [34,35,36]. Further studies are required to clarify which components of the investigated oils might be metabolized differently and which metabolite might be responsible for the different physiological responses, including hormonal and contractility changes. The lack of this information at this stage is a major limitation of our study. Our novel methodological approach, SMEMG, detects the contractile changes in living animals. Since the spectrum of myoelectric signals of the uterus and the cecum are in the same range and the cecum has a much larger muscle mass, it is very probable that we detected dominant cecum signals. It means that the changes in the non-pregnant uterine contractility can be monitored by the cecum SMEMG, which further aids the possible applicability of this method, including putative clinical use.

The study should be expanded to a larger number of rats or even to other species, with a special focus on fatty acid metabolism and cytokine profile studies in addition to the aforementioned methodological studies.

## 4. Materials and Methods

### 4.1. Housing and Handling of the Animals

Sexually mature (10–12 weeks old) female Sprague Dawley, Lister hooded, and Wistar rats (Animalab Ltd., Vác, Hungary) were used in the study. The animals were treated according to the Directive of the European Communities Council (2010/63/EU) and the Hungarian Act for the Protection of Animals in Research (Article 32 of Act XXVIII). All experiments involving animal subjects received prior approval from the Hungarian Ethical Committee for Animal Research (registration number: XIII./72/2020; date of approval: 25 February 2020). Considering the welfare of the animals, there were no exclusions during our studies based on the established exclusion criteria.

The rats were kept in an environment at a temperature of 22 ± 3 °C and a relative humidity of 30–70% with a 12-h light/12-h dark cycle. The standard rodent pellet diet (Altromin 1324 (SFA: 40.065 mg/kg, MUFA: 50.823 mg/kg, n-6 PUFA: 16.152 mg/kg, n-3 PUFA: 2.21 mg/kg), Animalab Hungary Ltd.) and tap water were provided ad libitum. The rats consumed an age-appropriate amount of food pellets, and there was no significant difference in the consumption of the different rats during the investigation. The alterations in the body weight of the rats were monitored through weekly measurements.

Each rat breed was divided into 3 groups and treated for 20 days with 1 mL of tap water (control), sunflower oil (SFO), or fish oil (FO) by gastric lavage. The daily and total dose of the different oils (Adexgo Ltd., Győr, Hungary) and their main components are provided in Table 1. Considering the 3Rs, the minimum experimental animal number was calculated using the Power and Sample Size program, based on α = 0.01 and 0.9 power settings, assuming equal numbers of control and treated individuals. Based on these, we needed to design 8 animals per control and 2 different experimental groups to exclude the null hypothesis (24 rats for each breed: in total, 72 rats). All animals were numbered and simultaneously randomized to the groups using a computer-based random order generator; then, the position of cages was also randomized. Investigators could not be blinded to the rat breed due to the difference in colors, but all animals in the experiment were handled, monitored, and treated in the same way.

### 4.2. Plasma Sample Collection and Hormone Analysis

On the first day of the study, 1 mL of blood samples was collected from the tail vein into tubes (143,458, BD Microtainer, Thermo Fisher Scientific Inc., Budapest, Hungary) containing K_2_EDTA (1 mg/tube) and subsequently centrifuged (1700× *g*, 10 min, 4 °C) to isolate plasma, and this procedure was repeated weekly to evaluate the effects of the various oil treatments. Plasma samples were preserved at −20 °C until hormone assays were performed.

Plasma concentrations of LH, progesterone (P4), and estradiol (E2) were quantified using an enzyme-linked immunosorbent assay (ELISA, ER1123, ER0492, and ER1507; Wuhan Fine Biotech Co., Ltd., Wuhan, China), following the protocols specified by the manufacturer. These kits are based on the competitive ELISA detection method. In total, 50 μL of the diluted plasma samples were added to each well with the biotin-labeled antibody and incubated at 37 °C. After incubation with HRP-Streptavidin and TMB solutions, the enzyme–substrate reaction was terminated by the addition of a sulfuric acid solution, and the color change was measured spectrophotometrically at a wavelength of 450 nm by a FLUOstar OPTIMA (BMG LABTECH, Ortenberg, Germany) microplate reader.

### 4.3. Smooth Muscle Electromyographic Measurements

Similar to blood sampling, SMEMG measurements were performed weekly. Food and water were withheld two hours prior to and during the measurement of myoelectric signals. Rats were anesthetized by isoflurane (AERRANE, Budapest, Hungary) inhalation (4% for induction, 2.5% for maintenance), and then a pair of bipolar disk electrodes was placed subcutaneously 1 cm below the midline above the abdominal cavity. SMEMG activity was recorded for a duration of 30 min using an online computer and amplifier system with the S.P.E.L. Advanced ISOSYS Data Acquisition System (MSB-MET Ltd., Veszprém, Hungary). After signal recording, the abdominal incision was sutured with a surgical thread, and the animals received a single 2 mg/kg subcutaneous injection of antibiotic gentamicin (Sandoz GmbH, Basel, Switzerland) to prevent infections. For another 2 days, a 5 mg/kg injection of analgesic ketoprofen (Ketofen, Merial, Lyon, France) was administered.

The SMEMG recordings from each group of rats were subjected to filtering for a frequency range of 1–3 cpm, which is indicative of both uterine and cecal activity and then analyzed using fast Fourier transformation [37,38]. Subsequently, the maximum power spectrum density (PsD_max_) values for 30-min intervals were statistically assessed. Due to the overlapping frequency range between the uterus and the cecum, in vitro contractility responses were also examined for the cecum for comparison.

### 4.4. Contractility Studies in Isolated Organ Baths

Laboratory animals were terminated using cardiac puncture under 4% isoflurane anesthesia immediately after imaging. The uterus and the cecum were excised, with fat and connective tissues removed, and subsequently rinsed with de Jongh solution for the uterus and Tyrode solution for the cecum.

The uterine and cecal samples were sectioned into 5-mm-long muscle rings; thereafter, these purified muscle strips were individually affixed to tissue holders and immediately placed in isolated organ bath chambers (IOB-08, MSB-MET Ltd., Hungary). In total, 10 mL of the appropriate buffer filled all chambers, and the temperature was maintained at 37 °C by continuous carbogen infusion (95% O_2_ + 5% CO_2_). Following mounting, the initial resting tension was set to 1.5 g, and the strips were allowed to equilibrate for 60 min with a buffer change every 15 min.

After the incubation period, spontaneous control contractions of the smooth muscle were induced by KCl (25 mM) [39]. The contractile activity of the tissue rings was recorded with a force/displacement sensor (FSG-01, MSB-MET Ltd., Hungary). For the purposes of recording and analysis, the S.P.E.L. Advanced ISOSYS Data Acquisition System (Figure 9) was used. The AUC was assessed at 5-min intervals, and the influence of KCl was expressed as a percentage of spontaneous contractions.

### 4.5. In Vivo Imaging Protocol

Rats were administered a highly selective imaging probe for αvβ3 integrin, IntegriSenseTM 680 (NEV10645, PerkinElmer Ltd., Springfield, IL, USA), which was diluted in phosphate-buffered saline and administered intravenously via the tail vein. An abdominal incision was made to facilitate live imaging under 4% isoflurane anesthesia. Imaging was executed 24 h post-injection using the IVIS Lumina LT III system (CLS136332, PerkinElmer Ltd., USA), employing 675 nm excitation and Cy5.5 emission filters, with an auto-exposure time and a binning factor of 2 on the final day of the experimental protocol. Fluorescence intensity was quantified using a multicolor scale ranging from red (indicating lower intensity) to yellow (indicating higher intensity). Fluorescence measurements were normalized to the size of the uterus and expressed as the average radiant efficiency ([photons/s/cm^2^/steradian]/[μW/cm^2^]) of the designated ROI.

### 4.6. Statistical Analysis

The levels of sex hormones, AUC, average radiant efficiency, and PsDmax values were calculated and compared after the parametric analysis of the dataset using an unpaired *t*-test. The *p*-values calculated through the unpaired *t*-tests, reflecting statistically significant differences, are shown in the appropriate figures. Statistical evaluations were performed using Prism 10 (GraphPad Software LLC, San Diego, CA, USA), with a significance level defined as *p* < 0.05.

## 5. Conclusions

The evidence we gathered is consistent with various studies published over the last few years. Consequently, two significant conclusions can be drawn: (1) within a single animal species, various breeds can exhibit different reactions to sex hormone status and smooth muscle contractions in response to PUFA supplementation, thus necessitating the preliminary assessments of the effects for each breed. (2) This consideration could be of paramount importance prior to both human nutritional investigations and the feeding of farm or laboratory animals, emphasizing the need to select the appropriate animal species and the specific breed within it.

In conclusion, the novelty of our findings is the proof of the breed-dependent effect of PUFA-containing oils on reproductive function. It is a widely recognized phenomenon that fertility rates among both human and agricultural animal populations in industrialized nations are currently on the decline. However, PUFAs are a double-edged sword; certain types are essential, but excessive intake may carry risks. Our understanding of the optimal balance of PUFAs required at various stages of life to achieve maximal fertility remains substantially limited, especially in relation to races of humans.

## Figures and Tables

**Figure 1 ijms-26-05221-f001:**
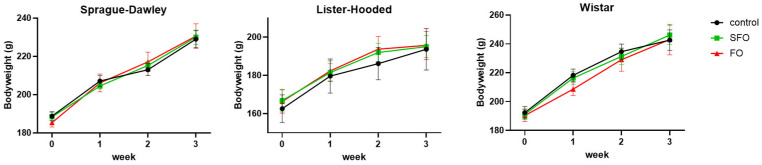
Increase in body weight (±SEM) of rats during the 3-week study. No significant changes were observed in any of the rat breeds in the case of any oil intake. SFO: sunflower oil; FO: fish oil.

**Figure 2 ijms-26-05221-f002:**
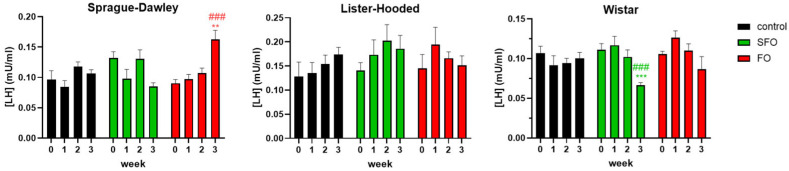
Breed-specific alterations in plasma luteinizing hormone (LH) concentration after different oil treatments. The LH level increased significantly with fish oil supplementation in the Sprague Dawley rats compared to the control group and the starting LH level of their own group; in contrast, a significant decrease was obtained for the Wistar rats in the group treated with sunflower oil, while there was no significant change in the Lister hooded rats. SFO: sunflower oil; FO: fish oil; **: *p* < 0.01; ***: *p* < 0.001 compared to the 3rd week value of the control group; ###: *p* < 0.001 compared to the initial LH level within their own group.

**Figure 3 ijms-26-05221-f003:**
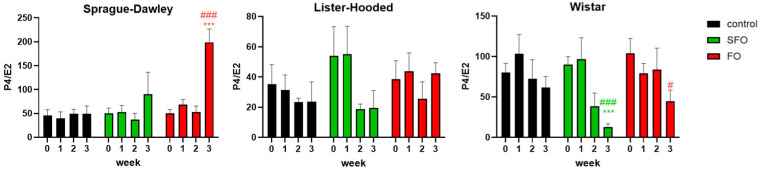
Changes in the P4/E2 ratio after different oil treatments. The sex hormone ratio was significantly increased by fish oil supplementation in the Sprague Dawley rats compared to the control group and the initial LH level of their own group; in contrast, a significant decrease was obtained for the Wistar rats in the sunflower and fish oil-treated groups, while there was no significant change in the Lister hooded rats. SFO: sunflower oil; FO: fish oil; ***: *p* < 0.001 compared to the 3rd-week value of the control group; #: *p* < 0.05; ###: *p* < 0.001 compared to the initial P4/E2 ratio within their own group.

**Figure 4 ijms-26-05221-f004:**
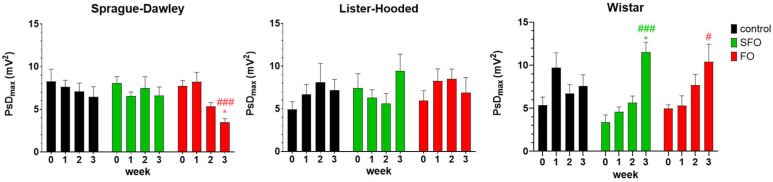
Changes in power spectrum density maximum (PsD_max_) values after 20-day oil treatment in anesthetized rats, detected by in vivo smooth muscle electromyography. Values are expressed as PsD_max_ ± SEM, showing the strength of muscle contraction. A significant decrease in the activity of spontaneous contractions was detected in the fish oil-treated Sprague Dawley rats compared to their control group; in contrast, a significant increase was obtained for the Wistar rats in both the sunflower and fish oil-treated groups, while there was no significant change in the Lister hooded rats. SFO: sunflower oil; FO: fish oil; *: *p* < 0.05 compared to the 3rd-week value of the control group; #: *p* < 0.05; ###: *p* < 0.001 compared to the initial PsD_max_ values within their own group.

**Figure 5 ijms-26-05221-f005:**
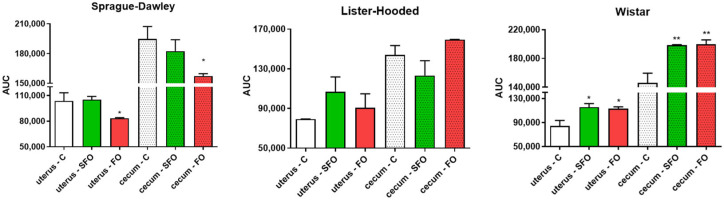
Effects of oil supplements on changes in spontaneous contractions of the non-pregnant uterus and the cecum. The change in contraction was evaluated through the area under the curve (AUC) ± SEM. A significant decrease was observed in the uterine and cecal samples from the Sprague Dawley rats in the fish oil-treated group; on the contrary, a significant increase was obtained for the Wistar rats in the sunflower oil-treated groups, while there was no significant change in the Lister hooded rats compared to the control groups. C: control; SFO: sunflower oil; FO: fish oil; *: *p* < 0.05. **: *p* < 0.01.

**Figure 6 ijms-26-05221-f006:**
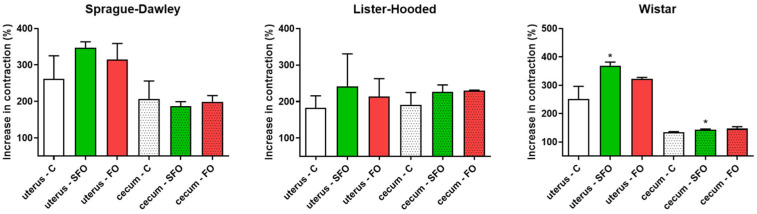
Effects of sunflower and fish oil treatment on changes in contractions of the non-pregnant uterus and the cecum. The relative change in contractions was obtained in each case compared to spontaneous contractions and was expressed as % ± SEM. A significant effect was observed only in the uterine and cecal samples of Wistar rats treated with sunflower oil compared to the control group. C: control; SFO: sunflower oil; FO: fish oil; *: *p* < 0.05.

**Figure 7 ijms-26-05221-f007:**
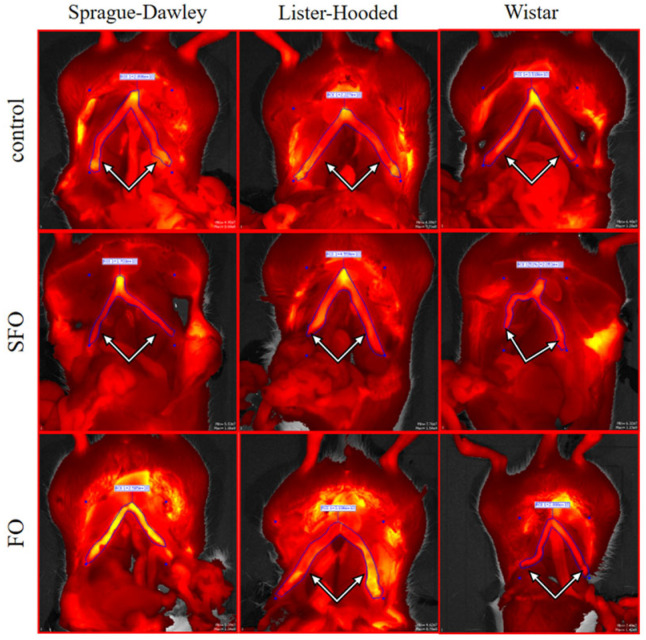
Representative fluorescent records of IntegriSense 680^®^ dye-labeled αvβ3 integrin in control, sunflower oil, and fish oil-treated non-pregnant rats. The yellowish intensity staining seen in the uterus indicates αvβ3 integrin, which is much brighter during the fertilization period. The white arrows indicate the uterine horns. SFO: sunflower oil; FO: fish oil.

**Figure 8 ijms-26-05221-f008:**
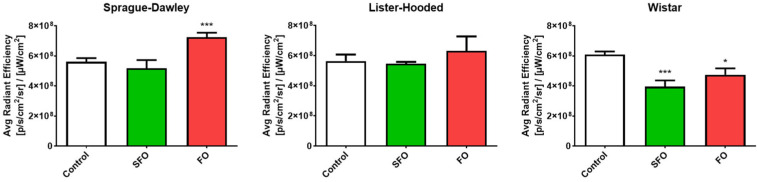
Average radiant efficiency (± SEM) in equally sized regions of interest (ROI) of the non-pregnant uterus for each rat breed. A significant increase in αvβ3 integrin was observed in the fish oil-treated Sprague Dawley rats, while in the case of the Wistar rats, treatments with sunflower and fish oil reduced the intensity compared to their own control group. We did not observe any changes due to treatments in the case of the Lister hooded rats. SFO: sunflower oil; FO: fish oil; *: *p* < 0.05; ***: *p* < 0.001.

**Figure 9 ijms-26-05221-f009:**
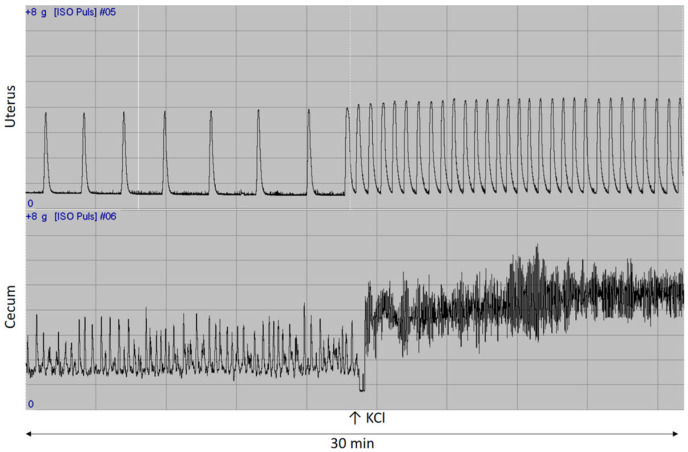
Representative 30-min records from the isolated organ bath study are presented. Spontaneous rhythmic contractions increased in the uterine rings (upper channel) and the cecum (lower channel) after the application of KCl (25 mM) derived from rats.

**Table 1 ijms-26-05221-t001:** Fatty acid content of sunflower oil and fish oil after a single daily dose and 20-day administration.

	SFO		FO	
	Daily Dose (mg)	Total Dose (g)	Daily Dose (mg)	Total Dose (g)
**SFA**	94.90	1.90	268.40	5.37
**MUFA**	302.50	6.05	351.80	7.04
**n-6 PUFAs**	521.70	10.43	61.40	1.23
**Total n-3 PUFAs**	0.83	0.02	238.50	4.71
**» ALA**	0.83	0.02	20.7	0.41
**» EPA**	-	-	85.38	1.71
**» DHA**	-	-	115.18	2.30

SFO: sunflower oil; FO: fish oil; SFA: saturated fatty acid; MUFA: monounsaturated fatty acid; PUFA: polyunsaturated fatty acids; ALA: α-linolenic acid; EPA: eicosapentaenoic acid; DHA: docosahexaenoic acid.

## Data Availability

The raw data supporting the conclusions of this article will be made available by the authors upon request.

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
