# Peer review of "Omega-3 Fatty Acid Consumption Alters Uterine Contraction: A Comparative Study on Different Breeds of Rats"

_ijms, 2025, doi:10.3390/ijms26115221_

Round 1

Reviewer 1 Report

Comments and Suggestions for Authors

GENERAL:

The MS "Omega-3 fatty acid consumption alters uterine contraction: A comparative study on different strains of rats" is in the scope of the journal with some significant results. The MS is well written, but there are some suggestions that should improve the MS.

ABSTRACT: Background:

Investigation of the effect of n-3 PUFA-rich fish oil and n-6-rich sunflower 14
oil on sex hormone status, in vivo and in vitro uterine contractility, and endometrial re-modeling. - uncomplete sentence - rewrite

INTRODUCTION

- clear; authors should highlight the novelty of this study

RESULTS

- well written

- for Figures 4, 5, 6 - also columns should be used as authors do not have exact data within the week

DISCUSSION

- some exact hypothesis in relation to strain-related effects should be added

- give clearly the novelty of the study and suggestions for further research

MATERIALS AND METHODS

- clear; suggestion: authors should implement an scheme describing the experiment with number of animals

CONCLUSIONS

- authors should highlight the novelty of this study

REFERENCES

- OK

Reviewer 2 Report

Comments and Suggestions for Authors

This manuscript is interesting because it studies the effect of fish oil- and sunflower oil-administered PUFAs on sex hormone levels and uterine muscle contraction in three strains of rats. It is interesting to note the different responses of these three rat strains to the administration oils.

The paper is generally well written, but cannot be accepted in the following form.

Please find my comments below.

MAJOR COMMENTS

- Since the three rat strains responded differently to the supplementation of the two types of oil, it would have been interesting to evaluate the biochemical profiles of the blood. Why was this not done?

- Lines 25-26: the sentence " in Wistar rats, both oils increased contractions" is incorrect because FO increased only the values of in vivo smooth muscle electromyography.

- References 7 in line 55 and 8,9 in line 58 do not seem appropriate to the context.

- To give a physiological approach to the work, describe the results following the sequence given in the Abstract (lines 15-16 and 22-24) and the Introduction (lines 101-102). The same sequence should be maintained in the Material and Methods section.

- Explain why the expression of αvβ3 integrin was evaluated.

- Discuss in detail the correlation between gonadotropic and sex hormone levels and the expression of αvβ3.

- Justify why cecal contraction was assessed.

MINOR COMMENTS

- Use the abbreviation when a molecule or method is used for the first time.

- Lines 107, 117, 129, 140, 150, 169, 182, 193: delete the dot after the number of the figure.

- Line 114: I would add ‘and SFO’ after ‘control’‘

- Delete brackets in lines 110, 124,125, 135, 146,148, 159, 161, 177,178, 187, 199, 200.

- Replace ‘Fig.’ with ‘Figure’ in the sections ‘Results’ and ‘Materials and Methods’.

- Line 120: do not write ‘through’ in italics.

Reviewer 3 Report

Comments and Suggestions for Authors

The work is very interesting: the extensive use of fatty acids in humans also necessitates a thorough understanding of the effects they have on various systems.

In this perspective, it's very important that the study is continued and expanded

Round 2

Reviewer 1 Report

Comments and Suggestions for Authors

As authors have done a serious review and all crucial comments were answered and/or revised the MS should be accepted for publication.

Reviewer 2 Report

Comments and Suggestions for Authors

The manuscript has been improved according to the suggestions of this reviewer and can be published in IJMS.